# Sampling bias adjustment for sparsely sampled satellite measurements applied to ACE-FTS carbonyl sulfide observations

Corinna Kloss[1,2], Marc von Hobe[1], Michael Höpfner[3], Kaley A. Walker[4], Martin Riese[1], Jörn Ungermann[1], Birgit Hassler[5], Stefanie Kremser[6], Greg E. Bodeker[6]

[1]Forschungszentrum Jülich GmbH, Institute of Energy and Climate Research (IEK-7), Jülich, Germany
[2]Laboratoire de Physique et Chimie de l'Environnement et de l'Espace (LPC2E), Université d'Orléans, CNRS, Orléans, France
[3]Karlsruhe Institut of Technology, Institute of Meteorology and Climate research, Karlsruhe, Germany
[4]University of Toronto, Department of Physics, Toronto, Ontario, Canada
[5]Deutsches Zentrum für Luft- und Raumfahrt (DLR), Institut für Physik der Atmosphäre, Oberpfaffenhofen, Germany
[6]Bodeker Scientific, Alexandra, New Zealand

*Correspondence to*: C. Kloss (Corinna.kloss@cnrs-orleans.fr)

**Abstract**. When computing climatological averages of atmospheric trace gas mixing ratios obtained from satellite-based measurements, sampling biases arise if data coverage is not uniform in space and time. Homogeneous spatio-temporal coverage is essentially impossible to achieve. Solar occultation measurements, by virtue of satellite orbits and the requirement of direct observation of the sun through the atmosphere, result in particularly sparse spatial
coverage. In this proof-of-concept study, a method is presented to adjust for such sampling biases when calculating climatological means. The method is demonstrated using carbonyl sulfide (OCS) measurements at 16 km altitude from the ACE-FTS (Atmospheric Chemistry Experiment Fourier Transform Spectrometer). At this altitude, OCS mixing ratios show a steep gradient between the poles and equator. ACE-FTS measurements, which are provided as vertically resolved profiles, and integrated stratospheric OCS columns are used in this study. The bias adjustment procedure requires no additional information other than the satellite data product itself. In particular, the method does not rely on atmospheric models with potentially unreliable transport
or chemistry parameterizations, and the results can be used uncompromised to test and validate such models. It is expected to be generally applicable when constructing climatologies of long-lived tracers from sparsely and heterogeneously sampled satellite measurements. In a first step of the adjustment procedure, a regression model is used to fit a 2-D surface to all available ACE-FTS OCS measurements as a function of day-of-year and latitude. The regression model fit is used to calculate an adjustment factor, which is then used to adjust each measurement individually. The mean of the adjusted measurement points of a chosen latitude range and season is then used as the bias-free climatological value. When applying the adjustment factor to
seasonal averages in 30° zones, the maximum spatio-temporal sampling bias adjustment was 11% for OCS mixing ratios at 16 km and 5% for the stratospheric OCS column. The adjustments were validated against the much denser and more homogeneous OCS data product from the limb-sounding MIPAS (Michelson Interferometer for Passive Atmospheric Sounding) instrument, and both the direction and sign of the adjustments were in agreement with the adjustment of the ACE-FTS data.

# 1    Introduction

Creating climatologies of atmospheric trace gas concentrations from satellite-based measurements is usually done by collecting available observations into latitudinal and monthly/seasonal bins and calculating the respective averages (e.g. Jones et al., 2012, and Koo et al., 2017, compiled comprehensive trace gas climatologies from ACE-FTS observations). For such methods, an evenly distributed coverage, with no significant measurement gaps, is
desirable to avoid introducing sampling biases when calculating climatological means. Satellite-based instruments, however, perform measurements only on distinct orbits, leaving spatio-temporal measurement gaps. This inhomogeneous sampling in space and time can introduce significant biases when calculating climatological averages (Aghedo et al., 2011;Toohey et al., 2013) if they are calculated in the traditional way. The magnitude of the sampling bias depends on the frequency spectrum of the spatial and temporal structure to be averaged. The bias can become particularly large when analysing data from solar occultation instruments that typically provide two measurements per orbit leading to sparse and spatially structured data coverage. The annual
solar occultation sampling pattern of ACE-FTS, is shown in Figure 1a.

Recent studies (Aghedo et al., 2011;Sofieva et al., 2014;Toohey et al., 2013;Millan et al., 2016) have investigated the effects of sampling biases for various satellite data products. Toohey et al. (2013) quantified the sampling bias for a number of satellites measuring ozone and water vapour. Depending on the trace gas, pressure level and latitude, they frequently found sampling biases as high as 20% and, in some cases, biases as high as 40% in regions with steep spatial and/or temporal gradients, such as in the vicinity of the polar vortex in both hemispheres. In an effort to quantify long-term trends in
stratospheric ozone between 60°N and 60°S, Damadeo et al. (2018) used a regression model (described in Damadeo et al., 2014) to estimate the sampling biases of several solar occultation instruments. They found that these biases lead to about 1% per decade absolute percentage differences in derived ozone trends. A common attribute of all previous methods used to estimate the sampling bias is that they either use additional/multiple data products or atmospheric models that use a priori knowledge of atmospheric transport and chemistry.

Here, we present a novel approach to adjust measurements to mitigate spatio-temporal sampling biases in climatological averages of carbonyl sulfide
(OCS) measured by the solar occultation instrument ACE-FTS. The method does not employ dynamical or chemical atmospheric models (e.g. CTMs) that may reflect inaccurate or incomplete understanding of the underlying processes. This approach thus allows the uncompromised application of the adjusted data product to test and validate such models.

The approach is suitable to be used on measurements with a seasonal cycle that is smooth enough to be represented by a low order expansion in Fourier series. Motivated by efforts to quantify the stratospheric burden of carbonyl sulfide (OCS) from ACE-FTS observations (Kloss, 2017), we use OCS
measurements from ACE-FTS. We introduce these measurements in Section 2 together with OCS measurements from Envisat-MIPAS that will be used to evaluate our method. Section 3 describes in detail the method developed to estimate and adjust for spatio-temporal sampling biases, which is then evaluated using the much denser and more homogeneous MIPAS data set in Section 4. Limitations of the method and its applicability to other tracers and regimes are discussed in Section 5.

## 2 Methods

### 2.1 ACE-FTS OCS observations

ACE-FTS is an infrared solar occultation spectrometer on the Canadian satellite SCISAT, delivering data since 2004 (Bernath et al., 2005). It measures in the spectral region from $750 – 4400$ cm$^{-1}$ ($2.2 – 13.3$ μm) with a spectral resolution of $0.02$ cm$^{-1}$. From these data, mixing ratio values are derived for over 30 trace gases together with temperature and pressure in selected altitude regions. As a solar occultation spectrometer, ACE-FTS retrieves only 30 profiles per day (two per orbit, at sunrise and sunset, with orbits spaced about 24° longitude apart) and thus exhibits significant data gaps in specific regions, as shown in Figure 1a. Measurements of the solar spectrum are made at tangent altitudes from 150 km down to 5 km (or cloud top) at a vertical resolution of 3 to 4 km. OCS mixing ratios are retrieved up to about 30 km altitude, above which the concentration typically drops below the detection limit.

Here, we use version 3.6 ACE-FTS OCS volume mixing ratio measurements between February 2004 and September 2016 (Boone et al. 2005, Boone et al. 2013), retrieved from microwindows in the range 2036 cm$^{-1}$ to 2056 cm$^{-1}$. The average fitting error for OCS is a statistical error for the retrieval from the fitting process and is between 1% and 3%, for the period considered here. A detailed analysis of OCS from ACE-FTS version 2.2 is presented in Barkley et al. (2008). Stratospheric OCS columns are calculated by vertically integrating concentration profiles from the dynamical tropopause to the top of the retrieved OCS profiles, where mixing ratios decrease to zero. The dynamical tropopause is defined as 380 K potential temperature in the tropics and 3.5 PV units at latitudes poleward of 30°, and is calculated from ECMWF ERA-Interim data (Dee et al., 2011).

When calculating climatological means of atmospheric trace gas mixing ratios at a given altitude, missing data over large parts of a region of interest do not automatically prohibit climatological averaging: an average can theoretically be created from one single data point, even though it may not be very representative of the true mean over the chosen spatio-temporal regime. Contrary, when calculating the stratospheric OCS burden over a particular latitude band and season, data coverage is critical irrespective of sampling bias because data have to be gridded and added up rather than being averaged. In our study, partial OCS columns are accumulated into 1° latitude bands over the chosen time period (e.g. one season: DJF, MAM, JJA, SON), and if there is more than one partial column in any bin, the mean is calculated. When adding up the burden for the chosen period, all 1° latitude bands have to contain realistic numbers, which is rarely the case with the sparse ACE-FTS sampling pattern. Therefore, bands with no profiles are either linearly interpolated from adjacent latitude bands or, close to the poles, linearly extrapolated from the two bands closest to the respective pole. If the gradient over the two bands closest to the poles is approximately representative for the gradient all the way to the pole, this procedure already accounts for potential sampling biases in a simplified way.

## 2.2 OCS observations by Envisat-MIPAS

The Michelson Interferometer for Passive Atmospheric Sounding (MIPAS) is a mid-infrared spectrometer on board the ESA (European Space Agency) satellite ENVISAT. It is a limb-sounding instrument, analysing the spectral radiance emitted by atmospheric trace gases. From its sun-synchronous polar orbit, MIPAS measures vertical profiles of multiple trace gases, including OCS. From 2002 – 2012 MIPAS operated in the spectral region between 685

– 2410 cm⁻¹ (4.1 - 14.6 μm), at a resolution of 0.025 cm⁻¹ until 2004 and then at 0.065 cm⁻¹ from 2005 onwards (Fischer et al., 2008). The vertical sampling is around 3 km in the altitude range from about 5 to 150 km above the clouds. With a horizontal sampling of about 400 to 500 km along the orbit MIPAS measured 1000 vertical profiles per day from 2002 to 2004 and 1400 between 2005 and 2012, covering almost all latitudes from 88°S to 88°N. These are about 40 times as many profiles as can be provided by ACE-FTS. OCS profiles are retrieved in spectral windows between 839 cm⁻¹ and 876 cm⁻¹ (Glatthor et al., 2015;Glatthor et al., 2017). The retrieval uncertainty for a single OCS scan is estimated to be 10% between 10 and 15 km, 26% at 20 km and

increasing up to 195% at 40 km altitude (Glatthor et al., 2015).

## 2.3 A regression model representation of the OCS field

Adjusting for spatio-temporal sampling biases requires some description of the gap-free field. The field could be obtained, for example, from chemistry-transport-model output, or, as mentioned above, from a satellite data set providing higher spatial and temporal sampling. In this study, we use the sparse

data themselves to create a gap-free OCS field through the application of a regression model fit. The regression model is used to fit a continuous smooth 2-D (time and latitude) surface either to OCS mixing ratios at a given altitude or to fields of OCS partial columns. In a general form with OCS represented by X, the regression model is:

$$X_{est} = a_0 + \sum_{i=1}^{N} \left[ a_{2i-1} \times \sin\left(\frac{2\pi d}{365.25}\right) + a_{2i} \times \cos\left(\frac{2\pi d}{365.25}\right) \right] \quad (1)$$

where the Fourier expansion in $N$ accounts for the annual cycle in the compound of interest and $d$ is the day of the year. To accommodate the

latitudinal structure in OCS, each of the $a_i$ coefficients are expanded in a Legendre series of index $M$. Values for $N$ and $M$ must be carefully selected to capture as much of the latitudinal and seasonal structure in OCS as possible, but must also avoid overfitting. For OCS, optimal fits were found for N=1 and M=4 resulting in a total of 15 fit coefficients. The output of Equation (1), $X_{est}$ is visualized in Figure 1 b. Applying fewer coefficients does not represent the OCS variability sufficiently, while applying more coefficients showed minima and maxima that are not observed in ACE-FTS as signs of overfitting.

A total of 12.5 years of ACE-FTS OCS mixing ratios at 16 km altitude are passed to the regression model to obtain the 15 fit coefficients (see Figure 1a). A different set of fit coefficients is obtained from the regression model when it is fitted to the stratospheric partial columns. Note that because the

regression model provides a value for any arbitrary latitude and day of the year, it meets the 'continuous' requirement for $X_{est}$. The extent to which the regression model can capture the true underlying morphology of the latitude vs. time OCS field depends on the OCS measurement coverage: however, with too many gaps in the measurements, the regression model will be required to have lower $N$ and $M$ expansions and may not capture subtleties in the OCS field to avoid over-and under fitting in areas of low data coverage. As a solar occultation spectrometer with only 30 measurements per day, ACE-

FTS exhibits significant data gaps in specific regions (as seen in Figure 1 a) that restrict the expansions in Equation (1) to N=1 and M=4.

This Fourier-Legendre fit only reflects the variability in the data with latitude and season that reoccurs every year. Using the entire 12-year data record for the fit yields the most robust result for this purpose. Any additional variability in the spatio-temporal pattern such as single events, trends, impact of El Niño, QBO, etc. is conserved, i.e. it will not be removed by the sampling bias correction. This might occur if the approximation was applied to each year individually.

The estimated regression fits for OCS mixing ratios at a given altitude or OCS partial columns describe the climatological, global state of OCS valid for the 12.5 years of available ACE-FTS observations. The coefficients for the regression fit are calculated by minimizing the sum of the squared differences between the original data (here the ACE-FTS observations) and the complete regression fit. This step in the regression is optimized by minimizing the differences simultaneously with respect to all coefficients used for the Fourier and Legendre expansions.

The regression model fit, together with its uncertainties, is therefore the best representation of the ACE measurements given the information provided

(original measurements, and number of Fourier and Legendre expansion settings) and due to the fitting process each fit coefficient has an associated uncertainty. Considering the uncertainties on the coefficients for the sampling bias adjustment process (see Section 3) would require the application of bootstrapping techniques to create many different realizations of the determined OCS climatologies to allow an estimate of the effects of the coefficient uncertainties on the determined sampling biases.

**3      Sampling Bias Adjustment**

Using the gap-free field as described in Section 2.3 ($X_{est}$), adjusted values can then be calculated as:

$$X_{adj} = X_{orig} \times \frac{\overline{X_{est}}}{X_{est}(lat, t)} \quad (2)$$

where X$_{adj}$ is the OCS value adjusted for its representativeness of the temporal-zonal mean, X$_{orig}$ is the unadjusted OCS measurement, $\overline{X_{est}}$ is some estimate of the true OCS temporal-zonal mean, and $X_{est}(lat, t)$ is the estimated OCS concentration at the location (note that only the latitude information affects the $X_{est}$ calculated by Equation 1) and time of the actual OCS measurement, sampled from the same source as $\overline{X_{est}}$. $t$ in Equation 2 only represents

season (day of year), so that there is only one combination of $X_{est}(lat, t)$ and $\overline{X_{est}}$ for any particular day of year and latitude. This is used to adjust

corresponding data points in every single year of the data set. Note that because the regression model provides a value for any arbitrary latitude and day of the year, it meets the 'continuous' requirement for $X_{est}$. $X_{est}$ does not have to be quantitatively correct – any biases divide out in Equation (2). There are several options for obtaining $X_{est}$. The only prerequisites are that the $X_{est}$ field represents the true underlying temporal and spatial morphology of the OSC field (though, as pointed out above, the values themselves do not need to be exact) and it needs to be continuous in so far as spatio-temporal means

can be calculated from the $X_{est}$ field without any spatio-temporal sampling gaps. The procedure for adjusting the sampling bias when calculating an average mixing ratio for a defined region over a given time period is illustrated in Figure 1. As examples, the method is explained in detail for two representative latitude-time boxes: one at 30 – 60°N for JJA (red box in Figure 1 a-c) and one for 60 to 90°S for DJF (black box in Figure 1a-c).

Figure 1 a shows the OCS mixing ratio values from 12.5 years of ACE-FTS observations as a function of latitude and time-of-year. The small year-to-year shifts in the latitudinal coverage of ACE-FTS causes small offsets between the traces for individual years seen in Figure 1 a. The red and black

boxes in Figure 1 indicate the selected time and latitude frames used to demonstrate the application of this method. The boxes were chosen as examples for the highest (red box) and lowest (black box) ACE-FTS latitude coverage. The climatological mean OCS pattern, represented as the regression model fit to the 12.5 years of ACE-FTS measurements, as a function of latitude and season, is shown in Figure 1b. Figure 2 shows the same for the OCS stratospheric columns. Values for $\overline{X_{est}}$ for the two example spatio-temporal means, indicated by the red (JJA, 30°N-60°N) and black (DJF, 60°S-90°S) boxes in Figure 1, can be calculated analytically, without any spatio-temporal sampling bias, from the regression model fit.

ACE-FTS data ($X_{orig}$) for 2010 are shown in Figure 1c. OCS mixing ratios from the regression model at the same latitudes and times as $X_{orig}$ provide $X_{est}(lat,t)$ allowing the original data to be adjusted using Equation (2). The advantage of applying Equation (2) rather than simply using $\overline{X_{est}}$ as the zonal mean seasonal mean is that trends and year-to-year variability observed in the data set are conserved. Equation (2) adjusts each measurement to be more indicative of the zonal seasonal mean. Figure 1d shows the adjusted ACE-FTS data set for the example of the red box in Figure 1c. These data points, now adjusted for their representativeness of the zonal seasonal mean, can then be used to calculate a better estimate of the true zonal seasonal mean for

the temporal and spatial domain of the red box. It should be noted that only derived averages are adjusted and not the individual data points. The average values should be more representative for the mean of the compound within each chosen box than without applying the adjustment method. The adjustment should not be applied to individual data points for any other purpose: clearly, the sampling bias is a systematic error type that only arises when deriving spatio-temporal averages and it does not impair the quality of individual data points at a particular location and time.

## 4    Evaluation of the adjustment procedure

**4.1    Case study results**

As seen in Figure 1a and shown in Barkley et al. (2008), OCS mixing ratios at a specific altitude (here 16km) decrease with increasing latitude. The stratospheric partial column distribution, shown in Figure 2, is quite different. Because both pressure and OCS mixing ratios rapidly decrease with height

above the tropopause, the major fraction of the stratospheric OCS column resides in the few kilometres just above the tropopause and thus the significant decrease in tropopause height with latitude leads to lower partial columns in the tropics and higher values closer to the poles. For the same reason, the annual cycle and day-to-day variability of the dynamical tropopause rather than the annual cycle in OCS mixing ratios largely controls the temporal variability of the stratospheric OCS partial columns, resulting in a more variable stratospheric OCS partial column field compared to the mixing ratio distribution shown in Figure 1a, potentially confounding the adjustment procedure.

Figure 3 shows the frequency distribution of ACE-FTS OCS measurements at 16 km from 2004 to 2016 for the two chosen latitude bands and time regions. The green histograms show the distribution of the original measurements and the blue histograms show the distribution of the adjusted measurements using Equation (2). Here, all individual measurements are adjusted for biases in the seasonal zonal mean. The shifts in the mean values and contraction of the standard deviations provide useful summary metrics of the effects of the applied spatio-temporal sampling bias adjustments. The distribution of all 12 years of data between 60°S and 90°S in the southern hemispheric summer (DJF) is shown in Figure 3a. This example was chosen because it displays the highest shift of 28 pptv or 11% in the mean OCS mixing ratios after applying the adjustment. The decrease in the mean value from 293 to 265 pptv in the latitude band from 60°S to 90°S can be explained by the fact that there are large measurement gaps at the southernmost latitudes, especially in DJF, and no measurements between 85°S and 90°S. Decreasing mixing ratios towards the poles, and measurement gaps where lower mixing ratios are expected, lead to a high biased mean over the chosen box (here: black box Figure 1a) when only averaging the available measurements. The true mean over the entire box is expected to be lower than the mean of only the available data. Thus, the shift of the mean to a lower value seen in Figure 3a qualitatively represents an adjustment of the simple data average towards the true mean of OCS mixing ratios over the entire box, and therefore at least a partial remedy for the sampling bias. Because Equation (2) generally shifts each data point towards the mean of the distribution, the standard deviation of the adjusted data will be lower than the standard deviation of the original data set. This is because in the original data set both measurement uncertainties and actual variability inside the considered box add on to the resulting standard deviation. Note that the observed reduction in the standard deviation (8 pptv in our black box example) reflects neither a reduction of the statistical uncertainty associated with the derived mean, nor a reduced variability over the entire box compared to only the available data. In fact, if actual observations covering the entire box were available, then their standard deviation would most likely be higher than that of the limited data because values would vary over a wider range of mixing ratios.

The histograms in Figure 3b show the data distribution for the red box in Figure 1, i.e. between 30°N and 60°N in northern hemispheric summer (JJA). Here, the adjustment method yields only a small shift in the average of 6 pptv (1.5%) because the entire chosen latitude range is covered by ACE-FTS measurements, which are therefore much more representative of the true mean value of the entire box compared to the previous example. For the red box, the original measurement values are more or less evenly distributed around the regression model mean, and Equation (2) shifts data towards the mean from both sides. Consequently, the reduction in the standard deviation by 32% is larger than in the previous example.

To assess whether the methodology quantitatively adjusts the sampling bias, a validation against an independent data set was performed and will be described in the following section.

## 4.2    A quantitative evaluation using MIPAS observations

To quantify the sampling bias arising from the sparse ACE-FTS sampling for a chosen latitude-time box, the OCS data product from the MIPAS instrument with its much denser data coverage is used. Because of the dense sampling pattern and almost complete latitude coverage (down to 88°S), the sampling bias of MIPAS is negligible compared to that of ACE-FTS. Figure 4 visualizes how much denser the MIPAS sampling is compared to that of
ACE-FTS globally (Figure 4 a) and in a chosen latitude-time box (Figure 4 b). Figure 4 a shows that both seasonal evolution and latitudinal variability of OCS mixing ratios at 16 km altitude are much better resolved by MIPAS than by ACE-FTS (c.f. Figure 1 a). Overall, seasonality and mixing ratio distribution agree well with the regression model output in Figure 1 b. Naturally, the regression tends to produce smoother gradients than the denser observations, e.g. the observed sharp decline in OCS at the southernmost latitudes in June (Figure 4 a) is smeared out in the regression (Figure 1 b). A notable difference between the MIPAS observations and the regression output is present at lower latitudes: while MIPAS OCS shows maximum values
in the subtropics around 30° in both hemispheres and a moderate local minimum in the tropics, the regression places the maximum close to the equator (with some seasonal variance) and shows decreasing OCS with latitude over all latitude ranges. The regression clearly inherits this behaviour from the individual ACE-traces shown in Figure 1 a, so this appears to be an instrumental difference between the MIPAS and ACE-FTS OCS data products. A systematic difference of 75 to 100 ppt lower OCS observed by ACE-FTS compared to MIPAS in the 14 to 20 km altitude region has been noted by Glatthor et al. (2017).

For the best possible quantitative evaluation, the spatio-temporal box in the ACE-FTS measurements with the lowest ACE-FTS coverage (Figure 4 b) and the highest observed sampling bias is chosen: December 2009 – February 2010, 60°S to 90°S (i.e. the black box in Figure 1). We compare the average of all MIPAS observations in a particular box to the average of only those MIPAS observations that are roughly equivalent in space and time to the available ACE-FTS observations in that box (i.e. only MIPAS measurements from 1st of December 2009 to 5th of January 2010 between 60°S and 68°S are used). Comparing all ACE-FTS and MIPAS measurement points between December 2009 and February 2010 in Figure 4 b again shows how much
denser the MIPAS sampling is compared to ACE-FTS. Like Glatthor et al. (2017), we also find the ACE-FTS mean value between 60°S and 90°S to be 115 ppt (28%) lower than the mean value of MIPAS. Therefore, relative rather than absolute mixing ratio differences are used to quantitatively describe the sampling bias in the comparison below.

Using the chosen spatio-temporal box (black box in Figure 1), we show in Figure 5 histograms of the relative frequency distributions of all MIPAS OCS mixing ratios at 16 km observed between 60°S and 90°S in DJF 2009/10 and of only those MIPAS observations roughly covering the ACE-FTS sampling
locations in that particular box (i.e. only MIPAS measurements from 1st of December 2009 to 5th of January 2010 between 60°S and 68°S are used). The histograms in Figure 3a and Figure 5 look similar in terms of shape and relative position. When we compare the two histograms in Figure 5 it becomes apparent that extending the sampling space over the entire box (down to 88°S) changes the distribution by adding additional lower mixing ratio values that were measured at the southernmost latitudes. The difference between the mean values of both histograms is 46 pptv, equivalent to a relative deviation

of about 11%, with the average of the full data set being lower. Thus, the difference has the same direction and magnitude as the shift in mean value when using the 'adjusted ACE-FTS data, compared to the original ACE-FTS data (Figure 3a). For this example, the performed sampling bias adjustment of the climatological mean from ACE-FTS data appears to work not only qualitatively but also quantitatively.

## 4.3    Significance

To investigate the scientific relevance and applicability of the proposed sampling bias adjustment, climatologies for the seasonal stratospheric OCS columns and OCS mixing ratios at 16 km altitude are calculated with and without sampling bias adjustments.

Due to the satellite orbit, ACE-FTS does not measure in the latitude ranges 85°N – 90°N and 85°S – 90°S, which can lead to a higher sampling bias close to the poles compared to the tropics and mid-latitudes where mostly all latitudes are covered within each season. Additionally, OCS mixing ratios exhibit lower stratospheric variability in the tropics. Therefore, the sampling bias is higher towards the poles and lower in the tropics. For the majority of points, from 60°N to 60°S (c.f. Figure 1), the modifications made using Equation (2) has only a minimal effect and is within the measurement uncertainty calculated using the ACE-FTS error estimates (see Toohey et al., 2010, for details on ACE-FTS error estimation).

The largest difference between the seasonal mean calculated using original OCS measurements and the seasonal mean calculated using the adjusted OCS measurements occurs in the latitude band 60°S – 90°S. Figure 6 shows the seasonal mean of the stratospheric column (top) and of mixing ratios at 16 km (middle) for this latitude band as calculated from the adjusted data set in red and the original ACE-FTS measurements in blue as well as the MIPAS mixing ratio equivalents (bottom). Due to the lower spatial coverage before 2008, only MIPAS data between 2008 and 2011 are considered. The relative difference between the mean values from the original and adjusted data set varies between 0.1 and 5.1% for the stratospheric columns (for the 5.1% difference: 1.29 kg/km$^2$ instead of 1.36 kg/km$^2$) and between 2 and 28 % for OCS mixing ratios at 16km. The largest adjustment of 28 % was observed in SON 2011, and unlike in virtually all other years and seasons, the mean mixing ratio was adjusted upwards from 195 ppt to 250 ppt. In 2011, the sampling of the 60°S – 90°S latitude band in the SON (shown in Figure 7 c) was even more sparse than in all other years (Figure 7 b) and the few valid data points are all located at the high latitude edge in the region where the regression model predicts lowest OCS mixing ratios (Figure 7 a). In addition, the OCS mixing ratios that were actually measured in SON were significantly higher in 2011 than in other years (compare Figure 7 b and c). The cause of these elevated OCS mixing ratios is currently unclear. The important thing to note in the context of our sampling bias correction is that the anomaly contained in the original data is conserved in the adjusted mean.

As described in Section 2.1, the procedure for the OCS stratospheric column integration already reduces the sampling bias by extrapolating OCS data into empty latitude bands. As a consequence, the sampling bias adjustment for the stratospheric burden is lower than for the mixing ratios. In this particular case (Figure 6), there is a marginal impact on the amplitude of the seasonal cycle as the adjustment most significantly reduces the austral summer OCS maximum at 16 km in virtually all years. No significant trends are apparent in either the original or adjusted data (-1.9 ± 2.3 · 10$^{-3}$ kg/km$^2$ per year for

the ACE-FTS stratospheric column and $-4.3 \pm 2.5 \cdot 10^{-3}$ kg/km$^2$ per year for the corrected column; $-0.2 \pm 0.8$ ppt per year for the ACE-FTS mixing ratios and $1.1 \pm 0.9$ ppt per year for the corrected mixing ratios; $0.05 \pm 0.43$ ppt per year for the MIPAS equivalent chosen according to the ACE-FTS sampling and $0.01 \pm 0.46$ ppt per year for the full mixing ration data set). Theoretically, if a sparse sampling pattern reoccurs each year (as for ACE), then the sampling bias does not affect long term (seasonal) trends but absolute climatological averages (such as the total burden). Trends related to dynamic changes in one particular region and season would also show up in both data sets if data from that region and season existed.

## 5    Conclusion and Discussion

In this study, we present a method to adjust the spatio-temporal sampling bias in climatologies calculated from sparsely sampled satellite observations without requiring additional observational evidence beyond the data set used. The fact that this method is exclusively based on observations and is independent of parameterization of atmospheric models makes it accessible for potential sampling-bias corrected climatologies used to test and improve such atmospheric models. Generally, the method can be applied to any atmospheric compound or property of which the variability follows defined seasonal and latitudinal patterns and can therefore be sufficiently well described using a regression model approach. The method has been shown to quantitatively adjust the sampling bias in seasonal 30° latitude band climatologies of OCS mixing ratios at 16 km altitude and OCS stratospheric column constructed from ACE-FTS observations. Our results show that, at least for OCS, the influence of the sampling bias is too small to significantly alter the scientific conclusions of climatological trends.

ACE-FTS with its solar occultation viewing geometry and therefore sparse and heterogeneous sampling pattern is particularly sensitive to the occurrence of a sampling bias when calculating climatologies (Toohey et al., 2013). OCS with its atmospheric variability in the stratosphere and upper troposphere limited to large spatial (100s of km) and temporal (i.e. seasons) scales (Barkley et al., 2008) provides an ideal tracer to investigate and demonstrate the sampling bias adjustment method. Note that the method would not work in the presented form (i.e. with a relatively simple regression model that is reasonably well determined by the data) for an OCS data product reflecting the lower tropospheric and boundary layer variability with complex regional patterns and to some extent distinct day-night differences such as IASI tropospheric OCS product described by Vincent and Dudhia (2017).

In the stratosphere and often in the UTLS, many long-lived trace gases (e.g. $N_2O$, CFCs) behave qualitatively similar to OCS with variabilities on similar scales. We expect the method to work well in the construction of climatologies for such tracers, explicitly including most compounds for which climatologies from ACE-FTS data have been compiled by Jones et al. (2012) and Koo et al. (2017). Toohey et al. (2013) addressed the sampling bias issue specifically for ozone ($O_3$) and water vapour ($H_2O$) measured by a wide range of satellites. Considering that the variability of both gases in the stratosphere and to a large extent the UTLS is dominated by distinct altitudinal, latitudinal and seasonal gradients, we expect a regression model such as the one described in Section 2.3 to adequately capture the largest part of this variability and, consequently, our sampling bias correction method to be applicable for both gases. Theoretically, with a denser satellite data product and a more elaborate version of the regression model that captures longitudinal

and other variabilities, the sampling bias correction scheme can be extended to climatologies that include other dependencies than just longitude and season. A detailed investigation and application of the method to $O_3$, $H_2O$ and other gases is beyond this 'proof of concept' study and remains to be investigated in the future.

## Acknowledgements

Measurements used in this study are from the ACE-FTS instrument and MIPAS together with the dynamical tropopause data from ECMWF. The Atmospheric Chemistry Experiment (ACE), also known as SCISAT, is a Canadian-led mission mainly supported by the Canadian Space Agency and the Natural Sciences and Engineering Research Council of Canada. MIPAS spectra used for deriving OCS vertical profiles at Karlsruhe Institute of Technology have been provided by the European Space Agency. The IMK/IAA-generated MIPAS data used in this study are available for registered users at http://www.imk-asf.kit.edu/english/308.php. Corinna Kloss has been supported by the graduate School of Forschungszentrum Jülich HITEC
(Helmholtz Interdisciplinary Doctoral Training in Energy and Climate Research) and ANR-17-CE01-0015 (TTL- Xing). Marc von Hobe was supported by the German Federal Ministry of Education and Research through the project ROMIC-SPITFIRE (BMBF-FKZ: 01LG1205C). The authors thank Kage Nesbit, Ben Lewis and Christian Rolf for their programming contribution.

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

**Figures**

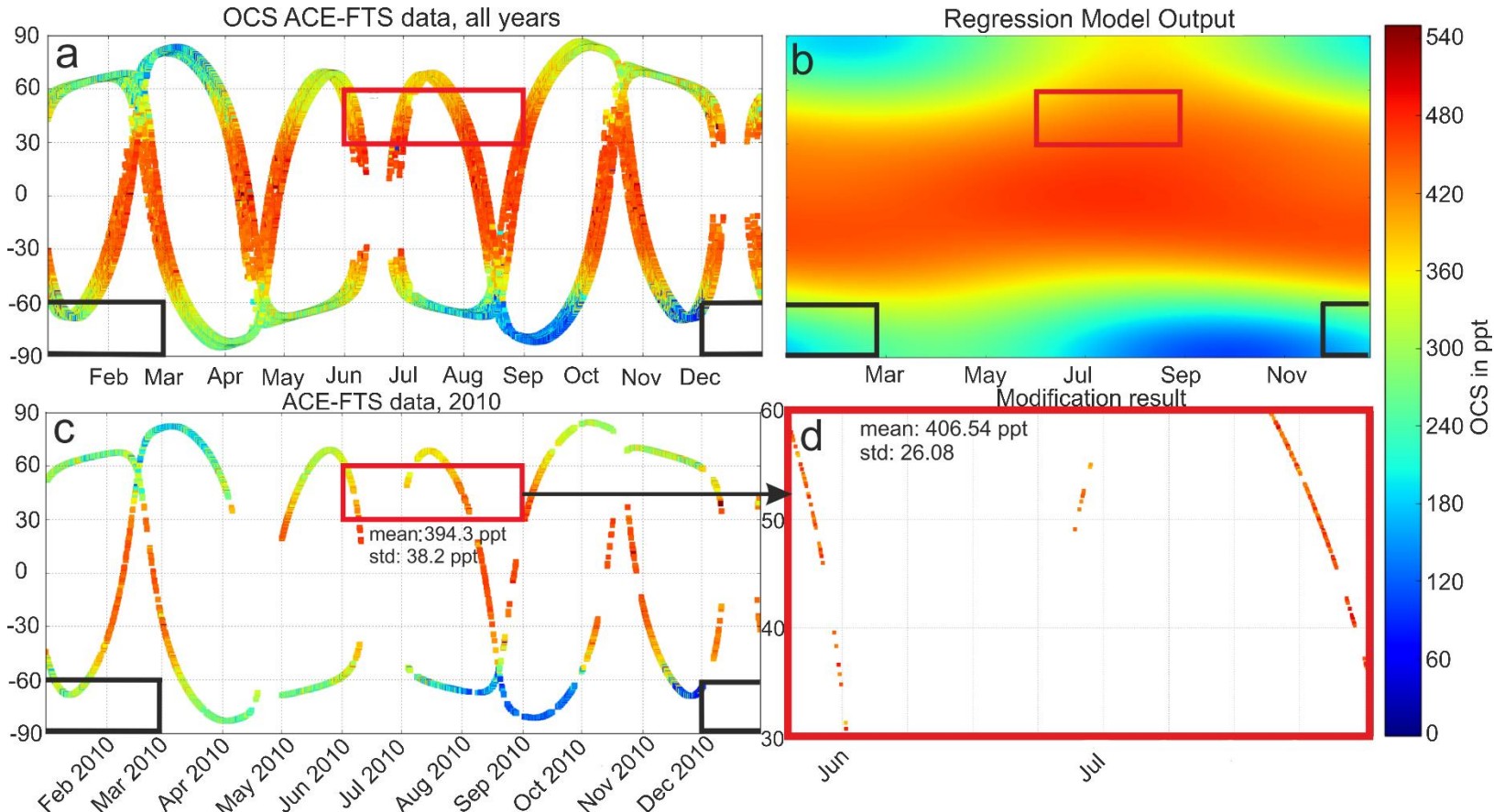

Figure 1: Schematic illustration of how the sampling bias is estimated and adjusted for OCS mean mixing ratio at 16 km altitude in any chosen time/latitude bin. Two examples are discussed in more detail in the text and are indicated by the red and black boxes. (a): All ACE-FTS measurements (2004 - 2016) as a function of day-of-year and latitude. (b): Regression model output to the ACE-FTS data of (a). (c) ACE-FTS measurements in 2010. (d): 'adjusted' data set, i.e. after the applying Equation (2) to the ACE-FTS measurements shown in (c), for the red box.

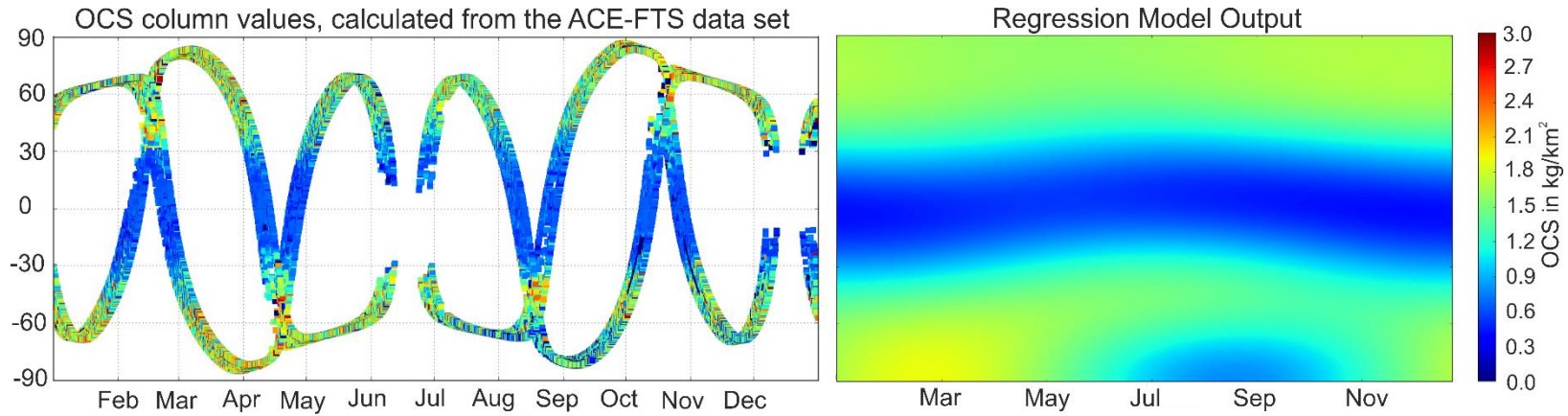

**Figure 2: Stratospheric OCS column values in kg/km$^2$ which were calculated for a 1° x 1° grid, using the ACE-FTS OCS data and the resulting Regression Model Output.**

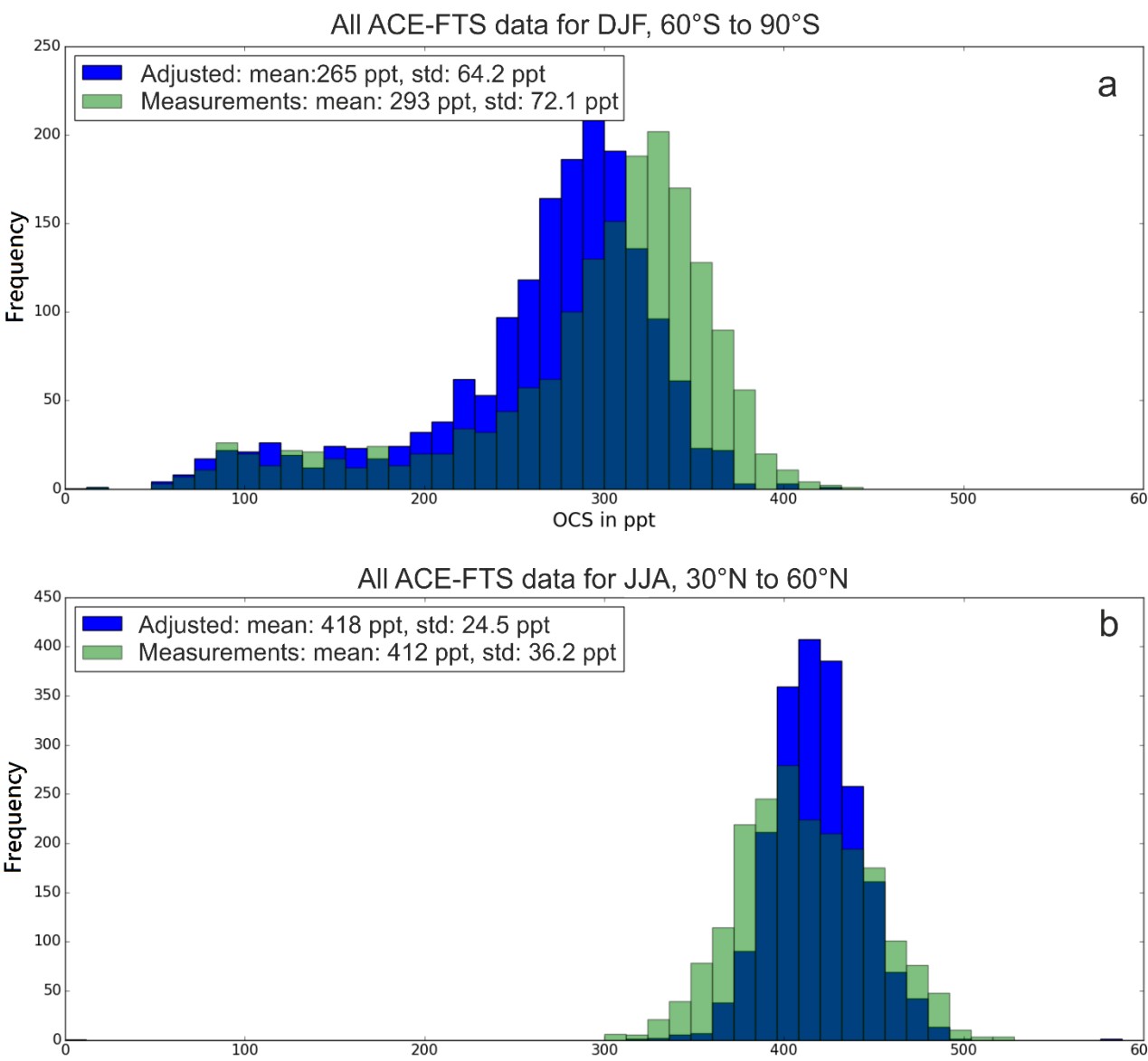

**Figure 3: Comparison of the distributions and resulting mean and standard deviation values of measured (green) OCS and the 'adjusted' measurements using Equation (2) (blue) for the same time/latitude bins indicated by the black (a) and red (b) boxes in Figure 1. Histograms include all 12 years of ACE-FTS OCS mixing ratio measurements at 16km altitude.**

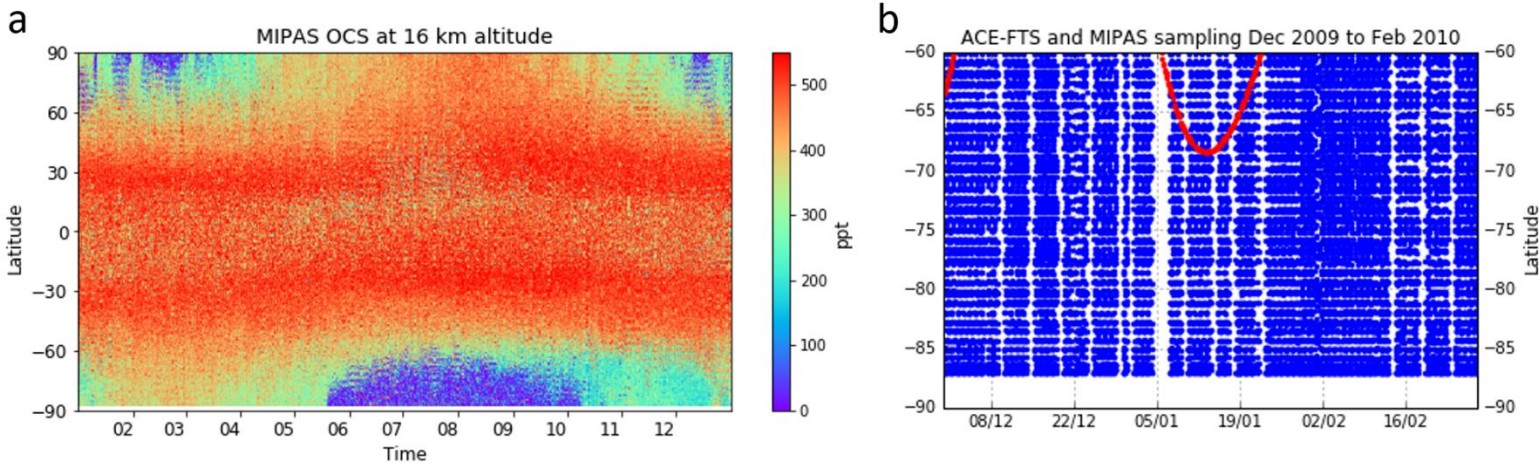

**Figure 4: The equivalent to Figure 1 a with the MIPAS OCS data set from 2008 to 2011 (a) and the sampling pattern of ACE-FTS in red and MIPAS in blue between 60°S and 90°S, December 2009 to February 2010 (b). The OCS colour scale is identical to that in Figure 1.**

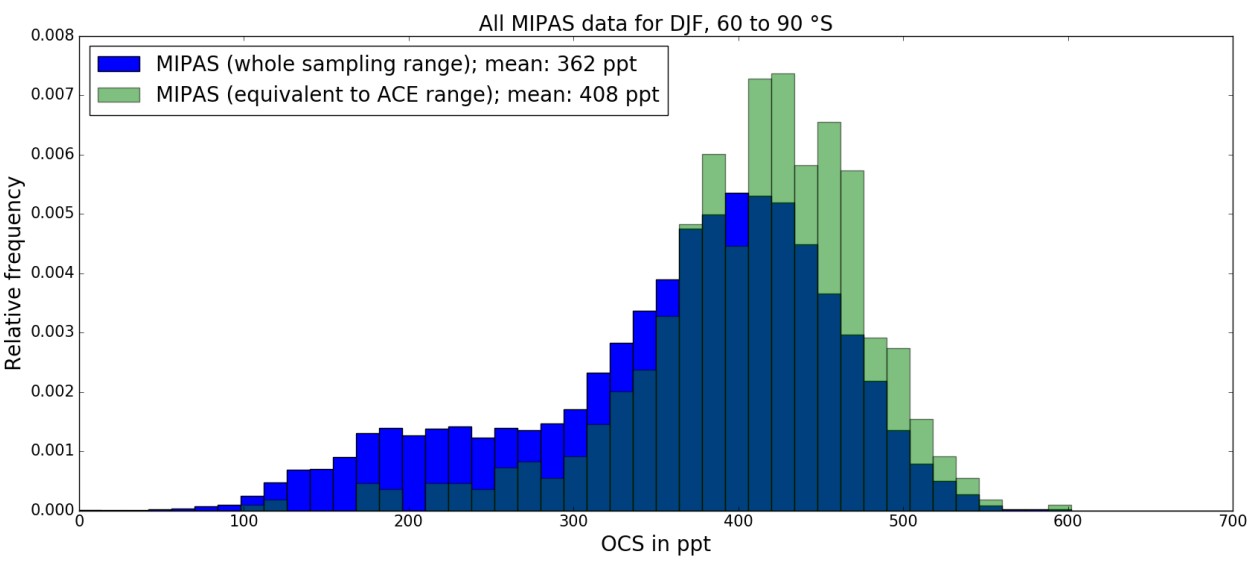

**Figure 5: MIPAS data distribution for DJF 2009 - 2010, 60°S to 90°S, for all available MIPAS OCS mixing ratio measurements at 16km altitude (blue) and for MIPAS OCS profiles in a comparable latitude and time frame as ACE-FTS measurements (green). The respective ACE-FTS plot, considering all years during DJF of ACE-FTS (to establish a reasonable statistic) is shown in Figure 3a.**

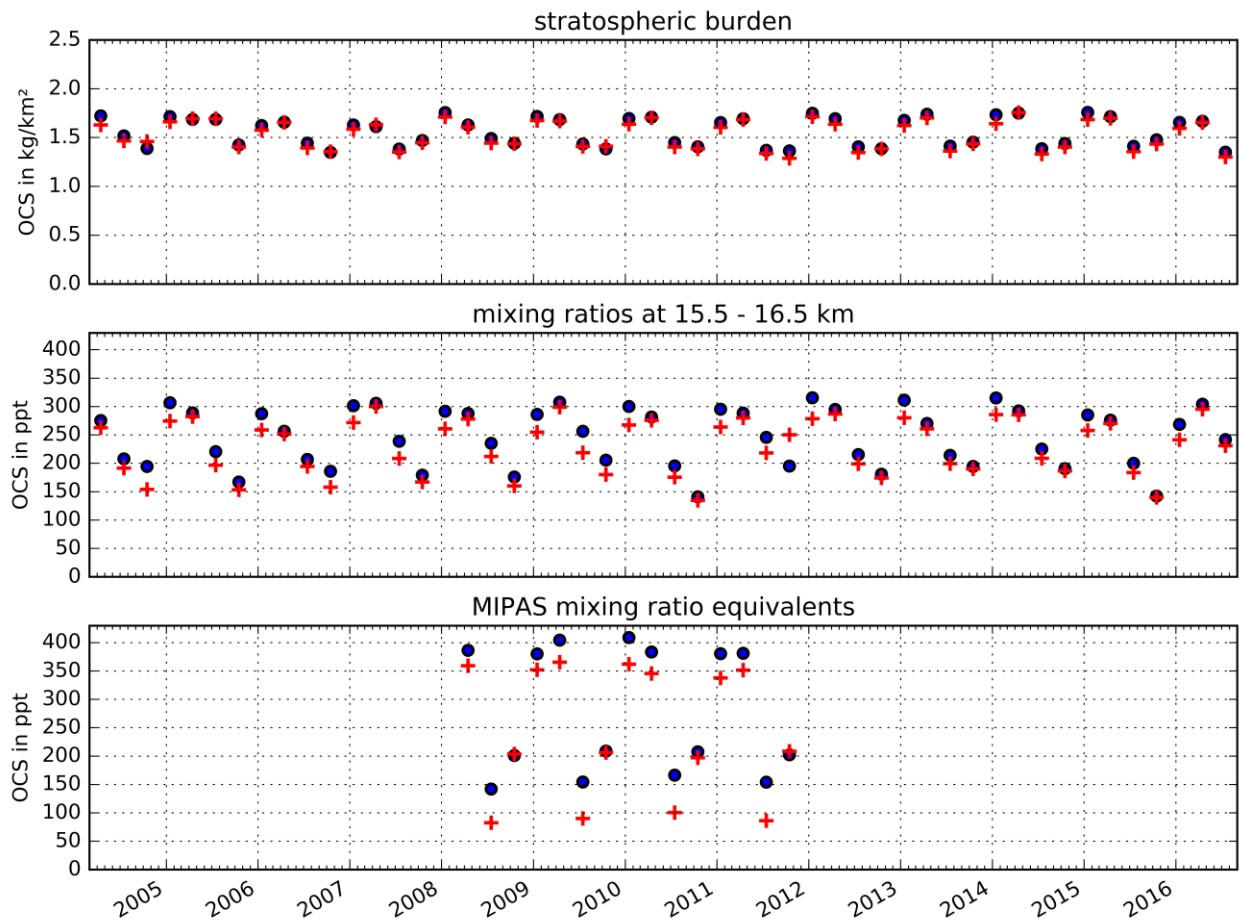

**Figure 6: Comparison of the unadjusted (blue) and adjusted (red) seasonal ACE-FTS OCS stratospheric columns and seasonal averaged OCS mixing ratio from 15.5 to 16.5 km altitude between 60°S to 90°S and MIPAS OCS mixing ratio equivalents.**

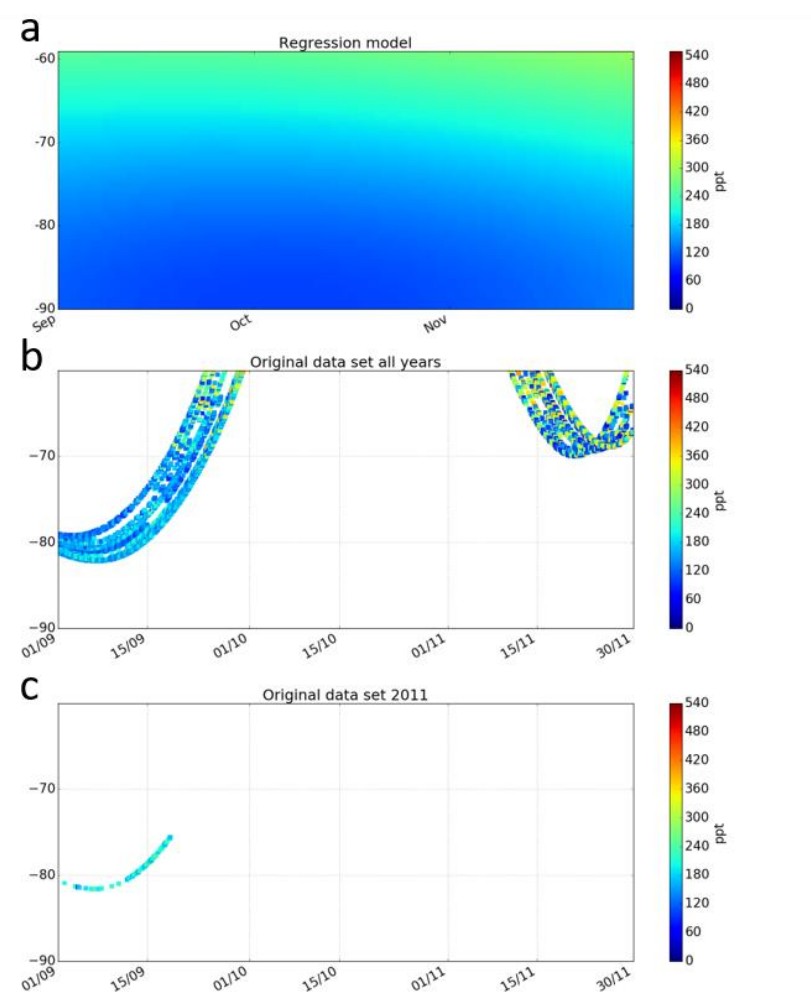

**Figure 7: Close-up of regression model (a) and actual ACE-FTS observations from all years (b) and 2011 (c) for the Sep-Nov season in the 60 – 90 °S latitude range.**