# Peer review of "Sampling bias adjustment for sparsely sampled satellite measurements applied to ACE-FTS carbonyl sulfide observations"

_Atmospheric Measurement Techniques, 2018_

## Referee Comment (RC1) · Anonymous Referee #1 · 1 Sep 2018

This manuscript describes a method to adjust OCS measurements from ACE-FTS to decrease the sampling bias when creating climatologies.

My main concern with the paper is the wider applicability of the method which the authors claim to discuss in section 5 (the conclusions section). However in that section, they only have the following sentence: "We expect the method to be applicable in the construction of climatologies for tracers with variabilities on similar scales, including most compounds for which climatologies from ACE-FTS data have been compiled by . . . "

To be acceptable for publication, the authors should showcase that their method works

for other molecules, like O3 and H2O (the molecules used by Toohey 2013), specially considering the next line in the conclusions: "Even though it is important to consider the sampling pattern of satellite based measurements, which leads to a sampling bias, at least for OCS the influence of the sampling bias is too small to significantly alter the scientific conclusions of climatologies." That is, the authors showcase their method to correct sampling biases in a molecule which does not have any significant sampling biases. The authors should either use model outputs sampled as performed in previous studies to prove the applicability of their method, or, construct O3 and H2O climatologies using all the available MIPAS measurements and those closest to the ACE-FTS measurement locations.

Specific comments:

Section 2.1 Mention the horizontal sampling of ACE-FTS for consistency with the MIPAS section.

Equation 2: where is the longitude information coming from. The model explained in section 2.1 is a 2D (time and latitude) regression.

Figure 2(right): Why is this regression model "Bodeker"

Figure 6: please use different symbols for the unadjusted and adjusted comparisons. Also bigger the symbols.

---

## Referee Comment (RC2) · Anonymous Referee #2 · 19 Oct 2018

The paper is dedicated to important issue: sampling bias adjustment of satellite measurements with a sparse sampling pattern, for the creating climatology. The method for the sampling bias correction based on regression model is developed and applied to the carbonyl sulphide measurements by ACE-FTS.

MAJOR COMMENTS

1) The assumptions of the applied method are not discussed sufficiently. The method uses an assumption that the spatio-temporal pattern can be represented by a smooth Fourier-Legendre expansion, the same for all years. This approximation is significantly simpler than, e.g., CTM simulations or a real evolution. Associated uncertainties should

be at least mentioned.

2) The advantages of the developed method are not demonstrated convincingly. In particular, evaluation using MIPAS data might be done in a more proper, from my point of view, way. In the present manuscript, the authors compare the histograms in 60-90S from 12 years on ACE-FTS measurements with those from 2 years of MIPAS measurements and conclude that the sampling bias correction improved climatology because histograms are similar. This is not very convincing, from my point of view.

A more proper way would be: select from the MIPAS measurements (the full dataset) a subset corresponding the ACE-measurements approximately, and apply the sampling bias adjustment described in the paper to this subset. Then compare the climatologies from the ACE-subsampled MIPAS dataset with and without sampling bias adjustment to the climatology from the full MIPAS dataset. Such evaluation would demonstrate the advantages and potential problems of the proposed method. In particular, the following can be studied/illustrated:

- Quantitative assessment of the sampling bias adjustment

- Changes in variability and trends (or their absence) due to sampling bias adjustment (see also minor comment #6)

- Changes in seasonal cycle

3) For evaluation of the method, it would be useful to compare the regression fit of ACE-FTS data shown in Fig 1 b with the analogous morphology using the MIPAS data.

4) The Section 4.3 , "Significance", with the first sentence starting with "To investigate the scientific relevance and applicability of the proposed sampling bias adjustment..." is expected to be a more deep analysis of the sampling bias correction. However, in the paper, the difference between adjusted and original datasets at 60-90°S are compared, without demonstration that the sampling adjustment improves the data record.

In particular, P.8, L.14: "There is a marginal impact on the amplitude of the seasonal

cycle" – Add quantitative values, please. Demonstrate that this is an improvement (by comparison with MIPAS, for example).

L. 15: "No significant trends are apparent in either the original or adjusted data" add quantitative estimates, please (here or in Fig. 6).

The statements at the end of the Sect 4.3. "Theoretically, …." related to changes in trends are not evident, especially taking into account that your sampling adjustment uses only on latitude and day of the year, i.e., it does not have a temporal dependence. The changes in trends need a more detailed analysis/discussion.

MINOR COMMENTS

1. Title of the paper needs a revision, from my point of view. It can be simply : "On sampling bias adjustment for sparsely observing satellite measurements", or "Sampling bias adjustment for sparsely observing satellite measurements with applications to ACE-FTS carbonyl sulphide measurements" or "On sampling bias adjustment for sparsely observing satellite measurements using regression modelling", or similar.

2. P.2., lines 18-19: Authors state: "To our knowledge, to date, no method has been reported where the quantification of a sampling bias, and the adjustments made to correct for it, does not require additional independent information."

It seems to be impossible to make a sampling bias correction without an additional information. It should not be confused here: the proposed method also relies on the additional assumption (information) that the spatio-temporal evolution can be developed into the regression model (Eq.1).

3. P.3, lines 16-17: "Partial columns are then accumulated into 1° x 1° bins over the chosen time period (e.g. one season: DJF, MAM, JJA, SON)". Is this the method for creating your climatology? Please clarify. Also the next sentence "Values for bins with no profiles are linearly interpolated or, close to the poles, are extrapolated from the two bins closest to the respective pole" seems to be in the contradiction with the method
described later in the paper. So, please describe the climatology in more detail: in particular, its spatial and temporal resolution, averaging method etc.

4. P.4, Eq.(1). I think it is worth to mention that you are characterizing the climatologic features only, not the temporal evolution.

Since you state that the method can be applied not only to OSC, I suggest changing variables in equations to more generic: i.e., Xest, Xorig etc.

5. P.5, Eq.(2) and the text: your model for fitting is the average in zones, while you refer to OSCest (lat, long, t). Please correct or explain.

6. P.5, lines 20-21: "The advantage of applying Equation (2) rather than simply using OSCEst as the zonal mean seasonal mean is that trends and year-to-year variability observed in the data set remain" This is not evident and needs to be shown. Since the sampling adjustment modifies the values, the trend and the variability can change. This can be evaluated/justified using the approach suggested in Major comment #2.

7. P.8, line 14. "Figure 5" Do you mean Figure 6 here?

8. P.9., last sentence: ". . .the influence of the sampling bias is too small to significantly alter the scientific conclusions of climatologies". What do you mean by "scientific conclusions of climatologies"?

9. P.13, Figure 2. I guess, the title of the right panel should be simply "Regression model output"
* * *

---

## Author Comment (AC1) · 25 Nov 2018

*The authors would like to thank Reviewer 1 for the constructive comments and concerns about the paper, which will help to improve the paper and make it more accessible. Below, each comment (black) is addressed (blue) in detail, indicating the changes we intend to make on the manuscript where applicable.*

My main concern with the paper is the wider applicability of the method which the authors claim to discuss in section 5 (the conclusions section). However in that section, they only have the following sentence: "We expect the method to be applicable in the construction of climatologies for tracers with variabilities on similar scales, including most compounds for which climatologies from ACE-FTS data have been compiled by . . . "

To be acceptable for publication, the authors should showcase that their method works for other molecules, like O3 and H2O (the molecules used by Toohey 2013), specially considering the next line in the conclusions: "Even though it is important to consider the sampling pattern of satellite based measurements, which leads to a sampling bias, at least for OCS the influence of the sampling bias is too small to significantly alter the scientific conclusions of climatologies." That is, the authors showcase their method to correct sampling biases in a molecule which does not have any significant sampling biases. The authors should either use model outputs sampled as performed in previous studies to prove the applicability of their method, or, construct O3 and H2O climatologies using all the available MIPAS measurements and those closest to the ACE-FTS measurement locations.

*Our paper is intended as a "proof of concept" for a new method to correct for sampling bias, and we feel that the comprehensive application to gases such as $O_3$ and $H_2O$ with all the potential scientific implications is well beyond that scope. Nevertheless, we agree with the reviewer that the one sentence in Section 5 falls short of a thorough discussion of the wider applicability, and that such a discussion is indeed warranted.*
*In the revised manuscript, we will provide arguments why the method should be applicable to a range of gases, and also point out some impediments that may arise with other gases. We will specifically address the examples $O_3$ and $H_2O$ in the light of what has been presented by Toohey et al. (2013) with respect to characterizing the sampling bias of these two gases.*

Section 2.1 Mention the horizontal sampling of ACE-FTS for consistency with the MIPAS section.
*We will add this information to the manuscript.*

Equation 2: where is the longitude information coming from. The model explained in section 2.1 is a 2D (time and latitude) regression.
*Theoretically, the model can work with longitude information. However, we are not using longitudes in our study and will therefore exclude the longitude dependency in the Equation.*

Figure 2(right): Why is this regression model "Bodeker"

*The reference to the model as "Bodeker regression model" is obsolete and will be removed from the paper.*

Figure 6: please use different symbols for the unadjusted and adjusted comparisons. Also bigger the symbols.

*Different and larger symbols will be used.*

---

## Author Comment (AC2) · 25 Nov 2018

*The authors would like to thank Reviewer 2 for the detailed and constructive criticism, comments and ideas. Below, each comment (black) is answered in blue. Where applicable, changes that will be made in the manuscript are indicated.*

**Major comments**

1) The assumptions of the applied method are not discussed sufficiently. The method uses an assumption that the spatio-temporal pattern can be represented by a smooth Fourier-Legendre expansion, the same for all years. This approximation is significantly simpler than, e.g., CTM simulations or a real evolution.

*Obviously, the reasons why we use a mathematical approximation rather than an atmospheric model have not come across in the current version. We will make sure to amend this in the revised version.*

*The major advantage of the mathematical fit is that it is based exclusively on the observations and is independent of any parameterizations in atmospheric models (e.g. CTMs) that may reflect inaccurate or incomplete understanding of the underlying processes. This becomes particularly important if sampling-bias corrected climatologies are later being used to test and improve such atmospheric models. Note that methods such as space-time spectra or Kalman filters to fit zonal patterns have been used in the past to transfer, for example, data from instruments such as LIMS (horizontal sounding of IR emissions) into level 3 data (synoptic maps). The combination of the data with a CTM would be data assimilation with the new data product inheriting the properties of the model, which is not what we want.*

*The Fourier-Legendre fit should reflect only the variability in the data with latitude and season that reoccurs every year, and using the entire 12-year data record for the fit yields the best statistics for this purpose. Any additional variability in the spatio-temporal pattern such as single events, trends, impact of El Ninho, QBO, etc. needs to be conserved, i.e. it should not be removed by the sampling bias correction. This would likely at least partly be done if the approximation was applied to each year individually.*

Associated uncertainties should be at least mentioned.

*We will add a discussion on the uncertainties of the regression model.*

2) The advantages of the developed method are not demonstrated convincingly. In particular, evaluation using MIPAS data might be done in a more proper, from my point of view, way. In the present manuscript, the authors compare the histograms in 60- 90S from 12 years on ACE-FTS measurements with those from 2 years of MIPAS measurements and conclude that the sampling bias correction improved climatology because histograms are similar. This is not very convincing, from my point of view.

A more proper way would be: select from the MIPAS measurements (the full dataset) a subset corresponding the ACE-measurements approximately, and apply the sampling bias adjustment described in the paper to this subset. Then compare the climatologies from the ACE-subsampled MIPAS dataset with and without sampling bias adjustment to the climatology from the full MIPAS dataset. Such evaluation would demonstrate the advantages and potential problems of the proposed method. In particular, the following can be studied/illustrated:

- Quantitative assessment of the sampling bias adjustment
- Changes in variability and trends (or their absence) due to sampling bias adjustment (see also minor comment #6)
- Changes in seasonal cycle
*We will include an analysis with the MIPAS data set from 2008-2011. The reduced time frame to 2008-2011 instead of starting from 2002 is due to the lower spatial coverage before 2008. We will add a Figure like Figure 6 with MIPAS equivalents.*

3) For evaluation of the method, it would be useful to compare the regression fit of ACE-FTS data shown in Fig 1 b with the analogous morphology using the MIPAS data.
*We will add another Figure (like Figure 1b) from MIPAS data without applying the regression model.*

4) The Section 4.3 , "Significance", with the first sentence starting with "To investigate the scientific relevance and applicability of the proposed sampling bias adjustment..." is expected to be a more deep analysis of the sampling bias correction. However, in the paper, the difference between adjusted and original datasets at 60-90S are compared, without demonstration that the sampling adjustment improves the data record.

In particular, P.8, L.14: "There is a marginal impact on the amplitude of the seasonal cycle" – Add quantitative values, please. Demonstrate that this is an improvement (by comparison with MIPAS, for example).
L. 15: "No significant trends are apparent in either the original or adjusted data" add quantitative estimates, please (here or in Fig. 6).
*Quantitative estimates will be added as requested.*

The statements at the end of the Sect 4.3. "Theoretically,...." related to changes in trends are not evident, especially taking into account that your sampling adjustment uses only on latitude and day of the year, i.e., it does not have a temporal dependence. The changes in trends need a more detailed analysis/discussion.
*We have revised this sentence as follows: 'if a sparse sampling pattern reoccurs each year (as for ACE), then the sampling bias does not affect long term (>> seasonal) trends but absolute climatological averages (such as the total burden).' In addition, the more detailed analysis with the MIPAS data set will help to rationalize this statement (see comment 2).*

**Minor comments**
1. Title of the paper needs a revision, from my point of view. It can be simply: "On sampling bias adjustment for sparsely observing satellite measurements", or "Sampling bias adjustment for sparsely observing satellite measurements with applications to ACE-FTS carbonyl sulphide measurements" or "On sampling bias adjustment for sparsely observing satellite measurements using regression modelling", or similar.
*As suggested, we change the title to 'Sampling bias adjustment for sparsely observing satellite measurements with applications to ACE-FTS carbonyl sulfide measurements'*

2. P.2., lines 18-19: Authors state: "To our knowledge, to date, no method has been reported where the quantification of a sampling bias, and the adjustments made to correct for it, does not require additional independent information."

It seems to be impossible to make a sampling bias correction without an additional information. It should not be confused here: the proposed method also relies on the additional assumption (information) that the spatio-temporal evolution can be developed into the regression model (Eq.1).
*We agree, therefore the sentence was deleted as requested. The information that the data set needs to fulfill certain criteria to be represented by a regression model follows two sentences later.*

3. P.3, lines 16-17: "Partial columns are then accumulated into 1°x 1° bins over the chosen time period (e.g. one season: DJF, MAM, JJA, SON)". Is this the method for creating your climatology? Please clarify. Also the next sentence "Values for bins with no profiles are linearly interpolated or, close to the poles, are extrapolated from the two bins closest to the respective pole" seems to be in the contradiction with the method described later in the paper. So, please describe the climatology in more detail: in particular, its spatial and temporal resolution, averaging method etc.
*Yes, it is the method for creating the climatology. However, this is not in contradiction to what is explained later on with the more advanced regression model. Here, a simple (as simple as possible, to show the 'before state') method to calculate OCS stratospheric burden values in a chosen region is explained. Later on, this method is represented by the red (unadjusted values) in Figure 6. For a better clarification, we change the paragraph: 'For the creation of a basic stratospheric column climatology with no advanced interpolation, partial columns are then accumulated into 1° x 1° bins over the chosen time period (e.g. one season: DJF, MAM, JJA, SON). Where there is more than one partial column in any bin, the mean is calculated. Values for bins with no profiles are linearly interpolated or, close to the poles, are linearly extrapolated from the two bins closest to the respective pole. To obtain the stratospheric burden for a particular region, respective columns are summed.'*

4. P.4, Eq.(1). I think it is worth to mention that you are characterizing the climatologic features only, not the temporal evolution.
*We will add such a statement in the revised manuscript.*

Since you state that the method can be applied not only to OSC, I suggest changing variables in equations to more generic: i.e., Xest, Xorig etc.
*We will change this in the manuscript.*

5. P.5, Eq.(2) and the text: your model for fitting is the average in zones, while you refer to OSCest (lat, long, t). Please correct or explain.
*Even though the model can potentially work with longitude information we are not using it and therefore we will remove this from the equation.*

6. P.5, lines 20-21: "The advantage of applying Equation (2) rather than simply using OSCEst as the zonal mean seasonal mean is that trends and year-to-year variability observed in the data set remain" This is not evident and needs to be shown. Since the sampling adjustment modifies the values, the trend and the variability can change. This can be evaluated/justified using the approach suggested in Major comment #2.
*We agree that this needs a clarification. We will explicitly emphasize that the "t" in Eq. 2 only represents season (day of year), and that the OCSest for any (lat,t) combination applied to correct the bias will be the same in every single year of the data set. Any variability between different years in the result will exclusively come from OCS orig. and is entirely conserved in the bias corrected product.*

7. P.8, line 14. "Figure 5" Do you mean Figure 6 here?
*Yes. This will be corrected.*

8. P.9., last sentence: "...the influence of the sampling bias is too small to significantly alter the scientific conclusions of climatologies". What do you mean by "scientific conclusions of climatologies"?
*To clarify, we changed this to 'the influence of the sampling bias is too small to significantly alter the scientific conclusions of climatological trends'*

9. P.13, Figure 2. I guess, the title of the right panel should be simply "Regression model output"
*That is correct and will be changed.*

---

## Author Response (AR1)

**Sampling bias adjustment for sparsely sampled satellite measurements applied to ACE-FTS carbonyl sulfide observations**

Corinna Kloss1,2, Marc von Hobe1, Michael Höpfner3, Kaley A. Walker4, Martin Riese1, Jörn Ungermann1, Birgit Hassler5, Stefanie Kremser6, Greg E. Bodeker6

[revised manuscript text omitted]